# Non-invasive ventilation with pursed lips breathing mode for patients with COPD and hypercapnic respiratory failure: A retrospective analysis

**Christoph Jünger**[1], **Maja Reimann**[2,3,4], **Lenka Krabbe**[1], **Karoline I. Gaede**[1,5,6], **Christoph Lange**[1,2,3,4,7], **Christian Herzmann** [1,2,5] *, **Stephan Rüller**[1]

1 Medical Clinic, Research Center Borstel, Leibniz Lung Center, Borstel, Germany, 2 German Center for Infection Research (DZIF), Borstel, Hamburg, Lübeck, Riems, Germany, 3 Division of Clinical Infectious Diseases, Research Centre Borstel, Borstel, Germany, 4 Respiratory Medicine & International Health, University of Lübeck, Lübeck, Germany, 5 Airway Research Center North (ARCN), German Center for Lung Research (DZL), Giessen, Germany, 6 BioMaterialBank Nord, Research Center Borstel - Leibniz Lung Center, Borstel, Germany, 7 Department of Medicine, Karolinska Institute, Stockholm, Sweden

* cherzmann@fz-borstel.de

**Data Availability Statement:** All relevant data are within the manuscript and its Supporting Information files.

## Abstract

### Purpose

Long-term non-invasive ventilation (NIV) is recommended for patients with stable chronic obstructive lung disease (COPD) and chronic hypercapnia. High inspiratory pressure NIV (hiNIV) and a significant reduction of arterial $pCO_2$ have been shown to prolong survival. Often, patients on hiNIV describe severe respiratory distress, known as "deventilation syndrome", after removal of the NIV mask in the morning. Mechanical pursed lips breathing ventilation (PLBV) is a new non-invasive ventilation mode that mimics the pressure-curve of pursed lips breathing during expiration. The clinical impact of switching patients from standard NIV to PLBV has not been studied so far.

### Patients and methods

In this hypothesis generating study, we retrospectively analysed the effects of switching COPD patients (stage GOLD III-IV) from conventional NIV to PLBV. Medical records of all patients who had an established NIV and were switched to PLBV between March 2016 and October 2017 were screened. Patients were included if they complained of shortness of breath on mask removal, used their conventional NIV regularly, and had a documented complete diagnostic workup including lung function testing, blood gas analysis and 6-minute walk test (6MWT) before and after 3–7 days of PLBV.

### Results

Six male and 10 female patients (median age 65.4 years; IQR 64.0–71.3) with a previous NIV treatment duration of 38 months (median; IQR 20–42) were analysed. After PLVB initiation, the median inspiratory ventilation pressure needed to maintain the capillary pre-switch

**Funding:** Fees for open access publishing were covered by a grant of the German Leibniz-Gemeinschaft, Open Access Publikation Fonds to Dr Christian Herzmann. The BioMaterialBank Nord is supported by the German Center for Lung Research. The BioMaterialBank Nord is a member of popgen 2.0 network (P2N). The funder had no role in study design, data collection and analysis, decision to publish, or preparation of the manuscript. The other authors received no specific funding for this work.

**Competing interests:** SR invented the PLBV and has co-developed the Vigaro within the scope of a consulting contract with FLO medical. CL reports personal fees from Chiesi, Gilead, Janssen, Lucane, Novartis, Oxoid, Berlin Chemie and Thermofisher outside the submitted work. CH reports personal fees from Janssen outside the submitted work. CJ, KIG and LK have no conflicting interests to declare. This does not alter our adherence to PLOS ONE policies on sharing data and materials.

**Abbreviations:** 6MWT, 6-minute walk test; AVAPS, average volume assured pressure support; BE, base excess; BiPAP, bilevel positive airway pressure; BMI, body mass index; CAT, COPD assessment test; COPD, chronic obstructive pulmonary disease; DGSM, German society for sleep research and sleep medicine; EIT, electrical impedance tomography; EPAP, expiratory positive airway pressure; FEV1, forced expiratory volume in one second; FVC, forced vital capacity; GOLD, Global Initiative for Chronic Obstructive Lung Disease; hiNIV, high intensity NIV; IPAP, inspiratory positive airway pressure; IQR, interquartile range; ITGV, intrathoracic gas volume; NIV, non-invasive ventilation; NT-proBNP, n-terminal pro brain natriuretic peptide; $pCO_2$, partial pressure of carbon dioxide; PCV, pressure controlled ventilation; PLBD, pursed lips breathing delay; PLBP, pursed lips breathing pressure; PLBV, pursed lips breathing ventilation; $pO_2$, partial pressure of oxygen; PSG, polysomnography; PT, pneumotachography; $R_{eff}$, effective respiratory resistance; RV, residual volume; SD, standard deviation; SOB, shortness of breath; TLC, total lung capacity; $VC_{in}$, inspiratory vital capacity.

$pCO_2$ level was reduced from 19.5 mbar (IQR 16.0–26.0) to 13.8 mbar (IQR 12.5–14.9; p<0.001). The median 6MWT distance increased from 200m (IQR 153.8–266.3) to 270m (IQR 211.3–323.8; p<0.001). Median forced vital capacity (FVC) increased from 49.5% to 53.0% of the predicted value (p = 0.04), while changes in $FEV_1$ and residual volume (RV) were non-significant.

## Conclusion

Based on this small retrospective analysis, we hypothesise that switching patients with COPD GOLD III-IV and chronic hypercapnia from conventional NIV to PLBV may increase exercise tolerance and FVC in the short term.

## Introduction

After non-invasive ventilation (NIV) was established in the 1980s for acute ventilatory failure, it is now also recommended as a long-term treatment for patients with stable chronic obstructive pulmonary diseases (COPD) and concomitant chronic hypercapnia in an outpatient setting [1]. This is based on numerous studies that demonstrated an improvement in the quality of life of patients with advanced COPD and stable hypercapnic respiratory failure [2–6]. In these patients, mortality rates can be reduced when high ventilation pressure is applied and a significant reduction in arterial $pCO_2$ is achieved [7]. A task force of the European Respiratory Society therefore suggests in a recent guideline, that long-term home NIV should be titrated to normalize or reduce $pCO_2$ levels in patients with COPD, but also acknowledges that the certainty of evidence for this recommendation is very low [8]. The term *high intensity non-invasive ventilation* (hiNIV) has been established for this ventilation method [9]. In patients with advanced COPD treated with NIV, the removal of the ventilation mask after nightly ventilation may result in shortness of breath, which has been described as *deventilation syndrome* [10]. A possible cause of the *deventilation syndrome* may be dynamic hyperinflation due to high inspiratory pressure of NIV.

Although high inspiratory pressures are applied in order to reduce hypercapnia, a recent study found no correlation between $pCO_2$ reduction and overall survival in this population [11]. This triggered a debate about the validity of chronic hypercapnia as the sole therapeutic indication for long-term NIV [11].

For the treatment of hypercapnia in patients with COPD, a variety of ventilation modes are used but no significant superiority of any mode has been demonstrated so far [12]. Since 2016, another ventilation mode is available through the NIV device "Vigaro" (FLO Medizintechnik GmbH, Melle, Germany). This device works in a time controlled or spontaneously triggered Bilevel Positive Airway Pressure (BiPAP) mode, but also mimics the pressure curve of spontaneous pursed lips breathing during the expiration phase (Fig 1).

This pressure curve results from two parameters to be adjusted in the device. The first parameter is called *pursed lips breathing delay* (PLBD) which determines the point in time at which the pursed lips breathing pressure starts after the onset of the expiration (range of delay: 0.0 to 0.4 sec). The second parameter is called *pursed lips breathing pressure* (PLBP), which determines the increase in pressure (range of altitude: 0.1 to 4.0 mbar) from the current pressure level. Typically, at low inspiratory positive airway pressure (IPAP) values, short PLBD and low PLBP are set, at higher IPAP values, longer PLBD and higher PLBP are set.

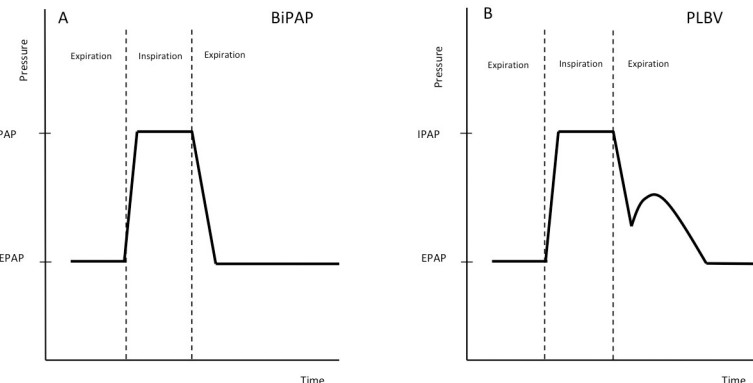

**Fig 1. Pressure and flow characteristics of BIPAP and PLBV ventilation modes.** Schematic pressure curve of a
BiPAP ventilation (A) compared to ventilation in PLBV mode (B). BiPAP: Bilevel positive airway pressure; IPAP:
inspiratory positive airway pressure; EPAP: expiratory positive airway pressure; PLBV: pursed lips breathing
ventilation.

After the PLBP is reached, the pressure decay is adjusted to the respiratory rate and reaches
EPAP in the last quarter of the expiration phase.

In this retrospective observational cohort study, we evaluated the effect of switching
patients with stable hypercapnic COPD and established NIV to PLBV on exercise tolerance
and lung function testing and hypothesized, that the minimization of NIV-induced hyperinfla-
tion by PLBV leads to an improvement of the clinical condition.

## Material and methods

This hypothesis generating, retrospective, monocentric analysis included consecutive patients
with hypercapnic COPD on NIV at the Medical Clinic, Research Center Borstel, Germany. All
patients with established NIV prior to hospital admission who were switched to PLBV between
March $1^{st}$, 2016 and October $31^{st}$, 2017 were screened for eligibility. Subjects were included in
the analysis if they had a diagnosis COPD GOLD stage III or IV without current exacerbation,
had a bilevel-NIV therapy that was established and regularly used for at least 4 months, com-
plained of shortness of breath on mask removal after their nightly ventilation, had evidence of
hypercapnia ($pCO_2 > 55$ mbar) in a capillary blood gas analysis obtained recently or prior to
NIV initiation, and had a complete diagnostic workup prior to and 3–7 days after PLBV appli-
cation documented in their medical records. This workup included a 6-minute walk test
(6MWT), nocturnal capillary blood gas analysis between 4.00 and 6.00 am under NIV, noctur-
nal polysomnography (PSG) including pneumotachography (PT) and spirometry with body-
plethysmography. The bodyplethysmograph "Masterscreen" (Jäger/CareFusion, Würzburg,
Germany) was used to measure the lung function. Polysomnographic controls were conducted
in a sleep laboratory certified according to the guidelines of the German Society for Sleep
Research and Sleep Medicine. Respiratory flow was measured with the pneumocontrol sensor
of a pneumoflow pressure transducer (ResMed, Martinsried, Germany) that registers flow sig-
nals on the principles of a pneumotachograph, but does not require calibration. "Rembrandt"
software (Medcare, Iceland) was used to evaluate the polysomnographic information that is
routinely used for the therapy adjustment of NIV. PSG allows the detection of AutoPEEP phe-
nomena, as described earlier for invasive ventilation (Fig 2) [13].

An AutoPEEP phenomenon was defined as a pneumotachographic signal due to an inspira-
tional effort of the patient without inspiration but instead reduction of the expirational airflow:

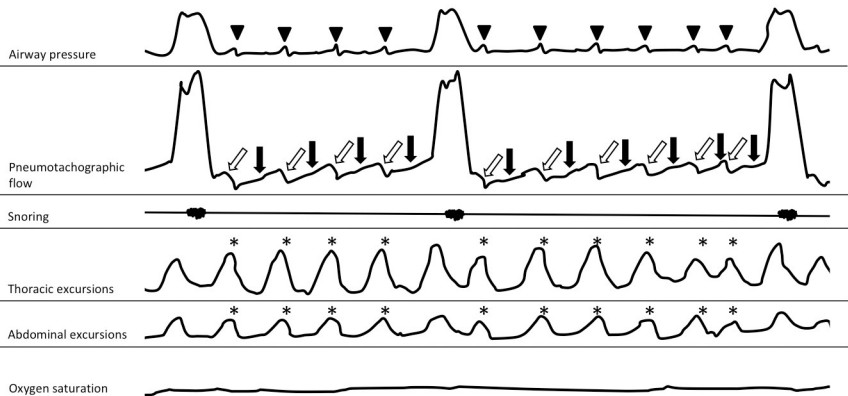

**Fig 2. AutoPEEP phenomena.** Schematic illustration of a severe AutoPEEP phenomena with repetitive AutoPEEP related frustrating inspirational efforts registered by polysomnography in a patient ventilated in an assisted mode. Inspirational efforts of the thorax and abdomen occur regularly (asterisk). No inspiration signal in the pneumotachographic channel is triggered, but expiratory flow is slowed (black arrows) and then accelerated (white arrows). The pressure channel registers a small peak (arrow head), resulting from a boosted expiratory flow.

In the late phase of expiration, it is often associated with a small short-term peak in the pressure channel and represents a boosted exhalation flow (Fig 2).

The settings of the PLBV were optimized to the lowest possible capillary $pCO_2$ at which no AutoPEEP phenomena occurred. A rising $pCO_2$ triggered an increase of IPAP. AutoPEEP phenomena triggered a decrease of IPAP. After patients were switched to PLBV, ventilation was monitored at night using polysomnography (PSG) including pneumotachography (PT) aiming to detect AutoPEEP (see Fig 2). If AutoPEEP phenomena were still present, the ventilation parameters were further optimized and monitored again by nightly PSG including PT. In most cases, a good NIV adjustment was achieved within 3–4 nights, rarely longer and once only after 7 nights. Follow-up investigations (lung function, blood gas analysis, 6MWT) were only performed after NIV was well adjusted.

COPD assessment test (CAT, GlaxoSmithKline Services Unlimited, Brentford, Middlesex, United Kingdom) as a routinely used surrogate parameter for disease-related quality of life was only available in 4 cases both before and after the NIV switch.

All patients included in this study agreed to the local broad consent of the BioMaterialBank Nord (University of Lübeck, Ethics committee AZ14-225). Use of patient related data for the present study was approved by the University of Lübeck, Ethics committee AZ 16–185).

## Statistical analysis

Data were analysed with the statistical program R version 3.5.1, open source software [14]. The Shapiro-Wilk-test was used to determine normal distribution. A paired sample t-test was used to compare variables whose difference between baseline and follow-up showed a normal distribution. The paired Wilcoxon signed-rank test was used as a non-parametric alternative. Effect sizes were calculated using the Cohens d method for normally distributed values and the Vargha & Delaney method for non-normally distributed parameters. For Cohens d, 0.5–0.8 indicates a medium effect size, >0.8 indicates a large effect size. For Vargha & Delaney, 0.64–0.71 indicates a medium effect size, >0.71 indicates a large effect size. Based on the effect sizes and an assumed power of 0.8 at a level of significance at 0.05 the following sample sizes were calculated: IPAP, n = 8 persons; 6MWT, n = 15 persons; FVC, n = 467 persons. Furthermore,

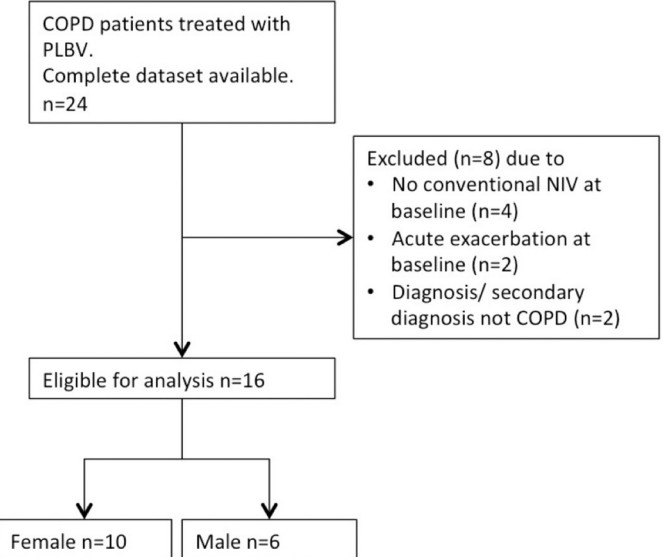

**Fig 3. Flowchart of the study.** Abbreviations: NIV, non-invasive ventilation.

ordinary least square regression (OLS) was used to test for correlations between the variables of interest and confounding factors.

## Results

Between March 1st, 2016 and October 31st, 2017, 24 patients were identified who received PLBV and had a full diagnostic data set available. From these, eight patients were excluded; two due to confounding pulmonary diagnoses (one with pulmonary fibrosis, one with α1-anti-trypsin deficiency), two due to an ongoing acute exacerbation of the COPD and four did not have well established or guideline-based home ventilation before PLBV (Fig 3).

Data were analysed from 16 patients (Table 1). 6 male (37.5%) 10 female (62.5%) with a median BMI of 25.8 kg/m$^2$ (Interquartile range, IQR 22.7–27.5). All patients lived in their own housing, took inhaled and oral medication in accordance with the current GOLD guidelines and were treated with long-term oxygen therapy, with a median of 2.0 L/min (IQR 1.8–2) that was continued throughout the hospital admission. Patients received NIV therapy for a median of 38 months prior to PLBV initiation (range 4–101 months). For n = 10 patients, categorical data describing exacerbation frequency during the preceding 12 months was available: 2

**Table 1. Patient characteristics.**

| Parameter | All patients (n = 16) |
|---|---|
| Age [years], median (IQR) | 65.4 (64.0–71.3) |
| Sex, n (%) | Male: 6 (37.5) Female: 10 (62.5) |
| BMI [kg/m$^2$], median (IQR) | 25.8 (22.7–27.5) |
| NT-proBNP [pg/ml], median (IQR) | 65.5 (55.0–160.0) |
| Duration of NIV therapy prior to PLBV [months], median (IQR). | 38 (20–42) |

Abbreviations: BMI, body mass index; NT-proBNP, n-terminal pro brain natriuretic peptide; IQR, interquartile range.

**Table 2. Ventilatory and clinical parameters at baseline and follow-up.**

| Parameter | Baseline | Follow up | p-value |
|---|---|---|---|
| **Ventilation** | | | |
| IPAP [mbar], median (IQR) | 19.5 (16.0–26.0) | 13.8 (12.5–14.9) | **<0.001***|
| EPAP [mbar], median (IQR) | 6.0 (4.8–9.0) | 6.0 (5.0–7.0) | 0.175 |
| **Oxygen supplementation** | | | |
| oxygen rate on NIV [L/min], median (IQR) | 2.0 (0.8–2.0) | 1.0 (0.9–2.0) | 0.722 |
| oxygen rate off NIV [L/min], median (IQR) | 2.0 (1.8–2) | 2.0 (0.8–2) | 0,371 |
| **Capillary blood gas analysis¶** | | | |
| pH, median (IQR) | 7.41 (7.40–7.43) | 7.41 (7.39–7.42) | 0.477 |
| $pCO_2$ [mmHg], median (IRQ) | 46.9 (43.8–55.7) | 50.0 (45.9–59.5) | 0,303 |
| $pO_2$ [mmHg], median (IRQ) | 76.5 (73.0–81.4) | 75.2 (70.7–81.0) | 0.713 |
| Base excess [mmol], median (IQR) | 5.2 (3.8–10.7) | 5.4 (3.5–10.2) | 0.900 |
| **Exercise tolerance** | | | |
| 6MWT distance [m], median (IQR) | 200 (153.8–266.3) | 270 (211.3–323.8) | **<0.001***|
| BORG scale before 6MWT, median (IQR) | 3 (1.5–3.75) | 3 (2.25–3) | 0.947 |
| BORG scale after 6MWT, median (IQR) | 7.5 (5.625–8.625) | 6 (5–7) | 0.191 |
| **Lung function tests** | | | |
| FEV1 [% predicted], median (IQR) | 24.5 (20.8–32.1) | 27.3 (20.8–31.3) | 0.203 |
| FVC [% predicted], median (IQR) | 49.5 (39.0–57.0) | 53.0 (41.8–60.8) | **0.040***|
| FVC [l], median (IQR) | 1.6 (1.3–2.1) | 1.8 (1.3–2.4) | **0.050***|
| FEV1/FVC [%], median (IQR) | 41.5 (35.1–47.4) | 38.6 (34.0–42.9) | 0.163 |
| $VC_{in}$[l], median (IQR) | 1.6 (1.3–2.3) | 2.0 (1.3–2.7) | 0.148 |
| TLC [% predicted), median (IQR) | 120.1 (116.8–149.3) | 140.0 (110.3–156) | 0.717 |
| ITGV [l], median (IQR) | 6.7 (5.5–7.3) | 6.6 (5.5–7.4) | 0.774 |
| $VC_{in}$/TLC, mean (IQR) | 0.3 (0.17–0.27) | 0.3 (0.21–0.32) | 0.177 |
| $R_{eff}$ [kPa×s×l$^{-1}$], median (IQR) | 1.1 (0.4–1.2) | 0.8 (0.7–1.1) | **0.027***|
| **Quality of life** | | | |
| CAT [score], median (IQR) | 26.5 (24.8–31.0) | 17 (11.3–22.5)§ | n.a. |

§At follow-up, only n = 4 patients competed the CAT questionnaire. Thus, no p-value is given.

¶Blood gas analysis was performed under oxygen supplementation in patients who had an established long-term oxygen therapy.

Abbreviations: IPAP, Inspiratory Positive Airway Pressure; EPAP, Expiratory Positive Airway Pressure; IQR, interquartile range; $pCO_2$, partial pressure of carbon dioxide; $pO_2$, partial pressure of oxygen; BE, Base Excess; 6MWT, 6 minute walk test: FEV1, Forced Expiratory Volume in One Second; FVC, Forced Vital Capacity; $VC_{in}$, inspiratory vital capacity; TLC, total lung capacity; ITGV, intrathoracic gas volume; $R_{eff}$, effective respiratory resistance.; CAT, COPD Assessment Test; n.a., not applicable.

patients (20%) had no exacerbation, 4 patients (40%) had 1–3 exacerbations and 4 patients (40%) had more than 3 exacerbations.

At baseline, the median IPAP was 19.5 mbar (IQR 16.0–26.0), while the EPAP was 6 mbar (IQR 4.8–9.0). As listed in Table 2, the median IPAP was significantly lowered to 13.8 mbar (IQR 12.5–14.9) (p<0.001, Cohens d 1.26), EPAP remained at 6.0 mbar (IQR 5.0–7.0) (p = 0.175). This pressure changes (ΔIPAP) led to various changes in $pCO_2$ (Δp$CO_2$) in some subjects, but did not result in a significant change of the median nocturnal $pCO_2$, with a median of 46.9 mmHg (IQR 43.8–55.7) at baseline and 50.0 mmHg (45.9–59.5) at follow-up (p = 0.303). Δp$CO_2$ for each patient is shown in Fig 4A. Fig 4B shows the correlation between ΔpCO2 and ΔIPAP (r = -0.21), which was not significant with β = -0.23 (p = 0.436). The

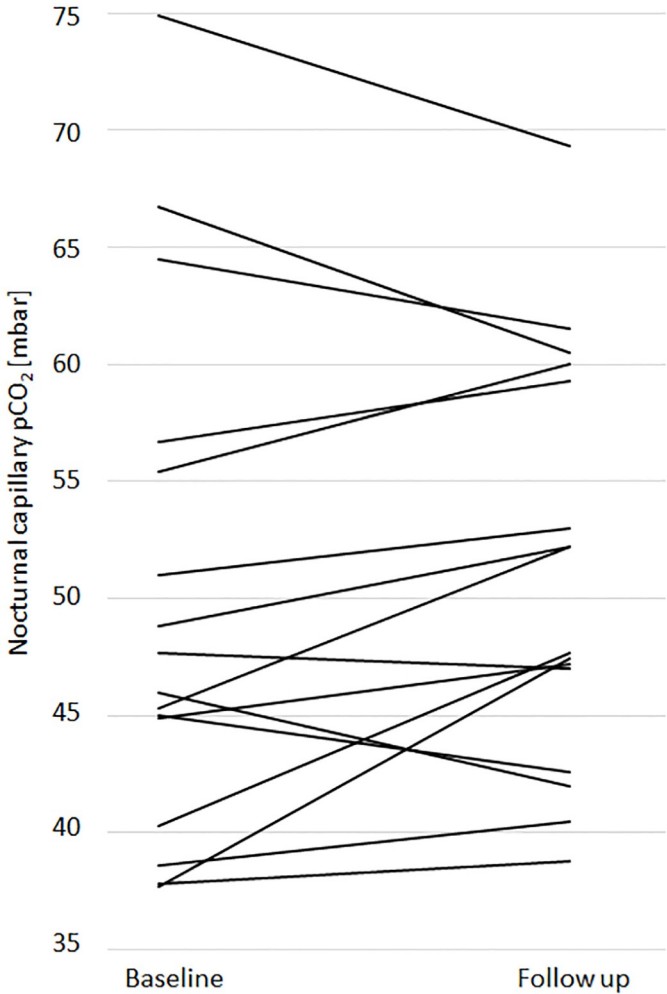

**Fig 4. pCO$_2$ values at baseline and follow-up and correlation of ΔpCO$_2$ with *ΔIPAP*. A.** Each line represents an early morning pCO$_2$ value for a single patient at baseline and follow up. **B.** Scatter plot illustrating the correlation of ΔpCO$_2$ (pCO$_2$ at follow-up—pCO$_2$ at baseline) and ΔIPAP (IPAP at follow-up—IPAP at baseline).

adjusted coefficient of determination of $R^2 = -0.02$ also underlines the lack of impact of ΔIPAP on ΔpCO$_2$.

The median walking distance in the 6MWT increased from 200m (IQR 153.8–266.3) to 270m (IQR 211.3–323.8) (Fig 5A). This median improvement of 70 m was highly significant (p<0.001, Cohens d 0.80). Patients reported less dyspnoea on exercise, although the changes in the BORG scale were not statistically significant. During the 6MWT, the median BORG scale increased from 3 to 7.5 prior to PLBV initiation and from 3 to 6 after the switch to PLBV as illustrated in Fig 6 (p = 0.19, Vargha & Delaney 0.64). As shown in Fig 5B, no correlation was found between ΔIPAP and the increase in walking test performance (r = 0.27). Increases in walking test performance of more than 100 meters were found for both small and large changes in IPAP. However, this effect was not significant with β = 3.33 (p = 0.304) and still seems to have a small effect on the variance of walking test performance difference (adjusted $R^2 = 0.01$).

And as shown in Fig 5C there was also no correlation between the extent of walking distance increase and pCO$_2$ changes (r = 0.04). OLS also confirms this missing correlation with a

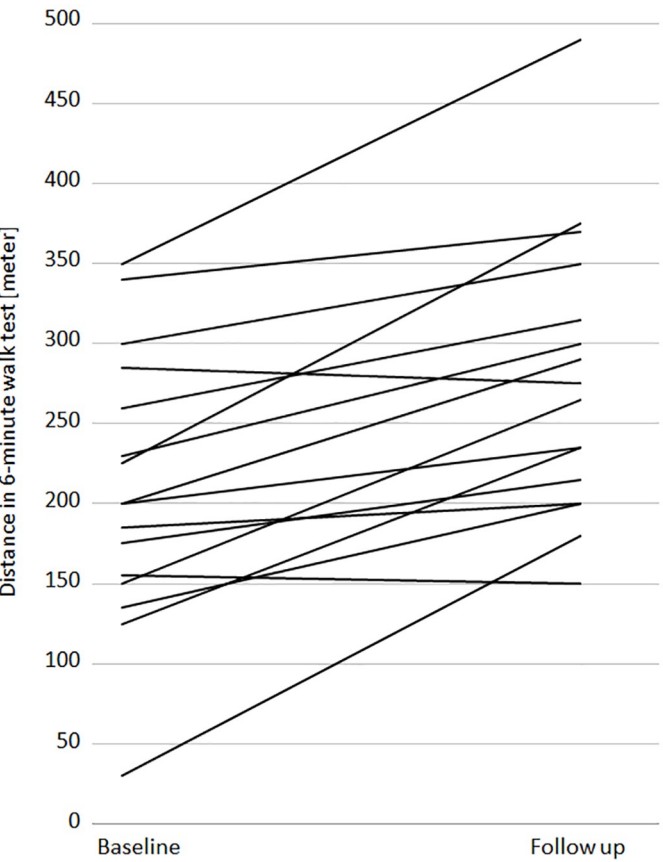

**Fig 5. 6MWT distances at baseline and follow-up *and correlations of Δ6MWT with ΔIPAP and ΔpCO₂*.** A. Walking distances in the 6MWT at baseline and at follow-up. Each line represents a patient. B. Scatter plot illustrating the correlation of Δ6MWT (6MWT at follow-up– 6MWT at baseline) and *ΔIPAP* (IPAP at follow-up—IPAP at baseline). C. Scatter plot illustrating the correlation of Δ6MWT (6MWT at follow.up– 6MWT at baseline) and *ΔpCO₂* (pCO₂ at follow-up—pCO₂ at baseline).

non-significant b = -0.41 (p = 0.894), and an explanation of variance of $\Delta pCO_2$ on the walking test performance is not given with an adjusted $R^2$ of -0.07.

Lung function showed a significant improvement in vital capacity (FVC) from median of 1.6 L (49.5%predicted) to 1.8 L (53.0%predicted) (p = 0.05 and p = 0.04, Cohens d 2.42). The improvement of the median effective airway resistance ($R_{eff}$) was also significant. It improved from 1.1 kPa*s*L$^{-1}$ to 0.8 kPa*s*L$^{-1}$ (p = 0.027, Vargha & Delaney 0.64). Other lung functional parameters showed a tendency but no significant change.

In a few patients (n = 4), data were available from the CAT questionnaires. In these patients the median result showed an improvement of 26.5 (IQR 24.8–31.0) to 17.0 (IQR 11.0–22.5) points (p = n.a.).

## Discussion

In this retrospective study, we describe quantifiable and clinically significant effects of switching COPD patients with shortness of breath on discontinuation of their nightly hiNIV to PLBV.

Most striking, the ventilation pressure decreased from a median of 19.5 mbar to 13.8 mbar without a significant increase in the median $pCO_2$. Although individual changes in $pCO_2$ were

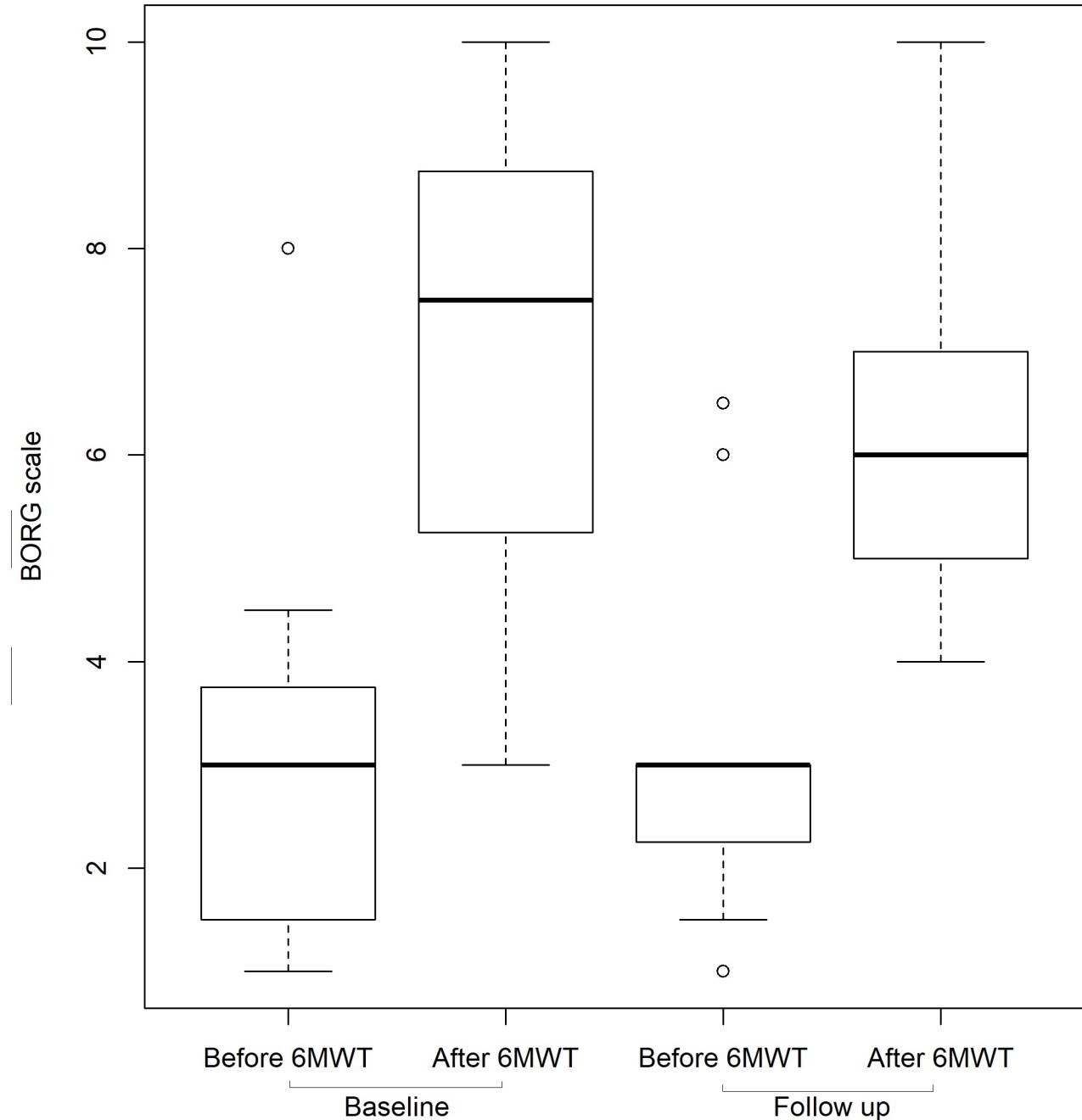

**Fig 6. Exercise induced dyspnoea at baseline and follow up.** Median BORG scale values before and after 6MWT. Baseline values were recorded before PLBV initiation; follow up values 3–7 days after PLBV initiation. Boxplots show median, interquartile range and range of dataset. Dots represent outliers. Abbreviations: 6MWT, 6 minutes walking test.

observed, these changes did neither correlate with the extent of pressure reduction nor with the result of the 6MWT. A plausible explanation for this phenomenon may be that PLBV enables deeper and longer expiration. This may lead to a lower end-expiratory intrathoracic gas volume (ITGV), improved diaphragmatic relaxation, and thus larger tidal volumes. Alternatively, PLBV improves the distribution pattern of ventilated air. Most likely, hiNIV leads to heterogeneous AutoPEEP and local areas of high PEEP associated with inadequate ventilation.

Since PLBV is striving to avoid AutoPEEP, these heterogeneously ventilated areas may be minimized. First attempts to support this hypothesis through the use of electrical impedance tomography (EIT) appear to augment this theory (data not published).

The main objective of PLBV was to avoid AutoPEEP phenomena. Nevertheless, the implementation of PLBV led to variable changes in capillary $pCO_2$, with a reduction in some patients, while others showed an increase or no change. Of note, in patients with higher initial $pCO_2$ values, PLBV often led to a $pCO_2$ reduction while an increase was seen in patients with lower initial $pCO_2$ values. One patient had an increase in $pCO_2$ of 7.4 mbar (from 40,3 to 47,7 mbar), another patient an increase of 9.7 mbar (from 37,7 to 47,4 mbar). Despite this laboratory finding, their 6MWT walking distance increased from 230 meters to 300 meters and from 150 meters to 265 meters, respectively. There was no correlation between $pCO_2$ and physical exercise tolerance in our cohort, suggesting an alternative mechanism.

Another striking result was the highly significant increase in 6-minute walking distance by 70 metres within less than a week after switching to PLBV. In two patients only, the 6MWT distance decreased after PLBV initiation. One patient discontinued the test due to knee pain. He walked 285 meters (BORG 7) before the PLBV, and 275 meters (BORG 5) after PLBV initiation. Oxygen therapy during NIV could be reduced from 2 L/min to 1 L/min. The other patient walked 155 meters before and 150 meters after PLBV-initiation. He had a pronounced corticosteroid associated myopathy as a consequence of long-term prednisolone use. In both walking tests, he took five breaks within six minutes. We assume that the myopathy was the performance limiting factor in the walking test.

This overall improved performance was accompanied by concordant spirometric function parameters that are congruent with other published data. Zikyri found a comparable improvement in FVC after initial initiation of outpatient NIV [4]. Nevertheless, the changes in lung function are on average only minor and do not reflect the clear differences in the 6MWT. It should be noted that all data originate from clinical routine and investigations were performed at varying times of the day. It is possible that body plethysmography was to detect more significant changes if performed immediately in the morning after mask removal. Additional bedside spirometry would add useful information and should be used in further studies.

There was also an impressive improvement in quality of life, based on the CAT questionnaire, which was only available in a few patients. However, the results seem plausible in view of the significant improvement in 6MWT.

In several patient files evaluated for this study, a clearly improved quality of sleep and a longer sleeping time were documented: fewer waking-up episodes due to nocturnal breathlessness, refreshing sleep and subjective well-being in the morning.

Before the transition to PLBV, COPD patients had considerable shortness of breath following mask removal in the morning. After conversion to PLBV, a considerable improvement of this *deventilation syndrome* was documented in patient files frequently. This finding is supported by a preliminary analysis of an ongoing prospective trial (NCT03299764) that showed a reduction of shortness of breath (SOB) from 8.8 (out of 10) points to 4.3 points on a visual analogue scale after 12 weeks PLBV in COPD patients [15].

Our study may have even underestimated the effects on exercise tolerance associated with PLBV. Several patients treated with conventional NIV experienced breathlessness so severe that they were not able to start a 6MWT or a pulmonary function examination because they did not tolerate the transport to the walking parcour or lung function department. Therefore, there was no recorded baseline data for these patients and they were not included in this study. Among them were two patients who were bed-ridden and NIV-dependent for 18 to more than 23 hours per day. After switching them to PLBV, NIV was only required at night and they were able to walk short distances (data not included).

Patients were more likely not to perform the 6MWT prior to the switch to PLBV than after PLBV initiation. Thus, excluding very weak patients from the analysis possibly led to an under-estimation of the PLBV effect. To minimize the selection bias, we decided to enrol only patients with a complete dataset (pre- and post-switch) into the study.

Since our results refer to retrospective, uncontrolled, non-blinded and observed data of a small group of patients, the conclusions drawn from the results are methodically limited. Given the small sample size, any additional patient enrolled may have had an impact on the study outcome. In the recruitment period, some patients were considered not to be suitable for PLBV by the treating physician; single patients did not accept the new ventilation method. It is impossible to avoid this subjective selection bias in a routine hospital setting when a new device is introduced since the selection criteria develop iteratively with growing experience. On the other hand, as described above, some patients were excluded due to their frail condition prior to PLBV initiation but would have been able to conduct all investigations some days later.

Furthermore, the algorithm driving the ventilation device was developed at our institution. Physicians who developed the algorithm were directly involved in patient care. This conflict of interest may have impaired an objective view on patient selection and treatment outcome. Potential conflicts of interest are declared in the supplement. Therefore, our observations can be used only to generate the hypothesis rather to confirm any clinical benefit of PLBV.

In order to minimize any potential bias, strict post-hoc inclusion and exclusion criteria were defined and only consecutive patients with complete data sets were included. Furthermore, we only assessed short term effects that were measurable within the clinical routine in the first week of PLBV initiation. A controlled prospective, randomised multicentre study is currently underway [16].

In conclusion, we found short-term improvements in exercise tolerance and FVC when patients with shortness of breath on discontinuation of their nightly hiNIV were switched to PLBV. These results have a limited validity as any retrospective analysis and based on a small number of cases but suggests the hypothesis that the new ventilation algorithm may have beneficial aspects for some COPD patients. Our findings are noteworthy as they suggest symptom improvements in COPD patients when their ventilation mode primarily aimed to avoid AutoPEEP. The findings from this small retrospective analysis warrant further investigations.

## Supporting information

**S1 Data.**
(XLSX)

## Author Contributions

**Conceptualization:** Christoph Jünger, Christoph Lange, Stephan Rüller.

**Data curation:** Christoph Jünger, Lenka Krabbe, Karoline I. Gaede, Christian Herzmann.

**Formal analysis:** Maja Reimann, Stephan Rüller.

**Funding acquisition:** Christoph Lange.

**Investigation:** Christoph Lange, Christian Herzmann, Stephan Rüller.

**Methodology:** Karoline I. Gaede, Christoph Lange, Stephan Rüller.

**Project administration:** Christoph Jünger, Lenka Krabbe, Christian Herzmann.

**Supervision:** Christoph Lange, Christian Herzmann.

**Visualization:** Christian Herzmann, Stephan Rüller.

**Writing – original draft:** Christoph Jünger, Stephan Rüller.

**Writing – review & editing:** Maja Reimann, Karoline I. Gaede, Christoph Lange, Christian Herzmann, Stephan Rüller.

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
