## [Decision Letter · Decision Letter 0]

7 Jul 2020

PONE-D-20-16968

Non-invasive ventilation with pursed lips breathing mode for patients with COPD and hypercapnic respiratory failure

PLOS ONE

Dear Dr. Herzmann,

Thank you for submitting your manuscript to PLOS ONE. After careful consideration, we feel that it has merit but does not fully meet PLOS ONE’s publication criteria as it currently stands. Therefore, we invite you to submit a revised version of the manuscript that addresses the points raised during the review process.

This is an interesting topic. The title of the article should be renamed to “Non-invasive ventilation with pursed lips breathing mode for patients with COPD and hypercapnic respiratory failure: a retrospective analysis”. It should be clearly stated in the Abstract and the Methods section that this is a hypothesis generating study. Although there is a Limitations paragraph in the Discussion section because of the several important limitations to the study this section should be discussed in significantly more detail together with how could this influence the results (sample size, selection bias, conflict of interest, etc.). In the light of what was said the conclusions should be rewritten.

We look forward to receiving your revised manuscript.

Kind regards,

Davor Plavec, MD, MSc, PhD, Prof.

Academic Editor

PLOS ONE

Additional Editor Comments:

This is an interesting topic. The title of the article should be renamed to “Non-invasive ventilation with pursed lips breathing mode for patients with COPD and hypercapnic respiratory failure: a retrospective analysis”. It should be clearly stated in the Abstract and the Methods section that this is a hypothesis generating study. Although there is a Limitations paragraph in the Discussion section because of the several important limitations to the study this section should be discussed in significantly more detail together with how could this influence the results (sample size, selection bias, conflict of interest, etc.). In the light of what was said the conclusions should be rewritten.

'SR: I have read the journal's policy and the authors of this manuscript have the following competing interests: I invented the PLBV and have co-developed the Vigaro device within the scope of a consulting contract with FLO medical.

CL: I have read the journal's policy and the authors of this manuscript have the following competing interests. I report personal fees from Chiesi, Gilead, Janssen, Lucane, Novartis, Oxoid,  Berlin Chemie and Thermofisher outside the submitted work.

CH: I have read the journal's policy and the authors of this manuscript have the following competing interests. I report personal fees from Janssen outside the submitted work.

CJ, KIG and LK: The authors have declared that no competing interests exist.'

a. Please confirm that this does not alter your adherence to all PLOS ONE policies on sharing data and materials, by including the following statement: "This does not alter our adherence to  PLOS ONE policies on sharing data and materials.” (as detailed online in our guide for authors http://journals.plos.org/plosone/s/competing-interests).  If there are restrictions on sharing of data and/or materials, please state these.

Please note that we cannot proceed with consideration of your article until this information has been declared.

4. Your ethics statement must appear in the Methods section of your manuscript. If your ethics statement is written in any section besides the Methods, please move it to the Methods section and delete it from any other section. Please also ensure that your ethics statement is included in your manuscript, as the ethics section of your online submission will not be published alongside your manuscript.

Reviewers' comments:

Reviewer's Responses to Questions

**Comments to the Author**

1. Is the manuscript technically sound, and do the data support the conclusions?

Reviewer #1: Yes

Reviewer #2: Yes

2. Has the statistical analysis been performed appropriately and rigorously? 

Reviewer #1: I Don't Know

Reviewer #2: I Don't Know

3. Have the authors made all data underlying the findings in their manuscript fully available?

Reviewer #1: Yes

Reviewer #2: Yes

4. Is the manuscript presented in an intelligible fashion and written in standard English?

Reviewer #1: Yes

Reviewer #2: Yes

5. Review Comments to the Author

Reviewer #1: Authors have good knowledge of physiological process of breathing,

Their idea of applying another mode of NIV similar to the well established exercise of breathing with pursed lips and its implementation into praxis in problematic COPD patients is very fresh and promising.

The problem is very low number of investigated subjects and the retrospective manner of study. In these circumstances I do not feel myself enough competetent to assess statistical analysis.

Some results are unexpected, it is difficult to explain how the pCO2 was not lower with such improvement of exercise capacity measured by 6MWT. The only possibility that drop on my mind is that the measurement of pCO2 in capillary sample is not accurate enough, and that arterial sample is more appropriate, so in hypothetical next similar study it should be implemented.

All results show that the new method PLBV decrease hyperinflation, which contribute significantly to the quality of life, what is one of the most important goals in management of COPD, thus putting the PLBV on the important place in treatment of severe COPD patients.

Reviewer #2: Interesting results of the retrospective analyse, prompt the prospective study. Nothing major to correct. Accent to the quality of life and exercise tolerance over the lung function and blood gasses in COPD management.

6. PLOS authors have the option to publish the peer review history of their article (what does this mean?). If published, this will include your full peer review and any attached files.

Reviewer #1: **Yes: **Sanja Popović-Grle

Reviewer #2: No

---

## [Author Response · Author response to Decision Letter 0]

22 Jul 2020

Dear Editor,

please find enclosed the revised manuscript as suggested in your email dated July 13th, 2020.

As suggested, the following amendments were made:

1) The Competing Interests section now reads as follows:

SR invented the PLBV and has co-developed the Vigaro within the scope of a consulting contract with FLO medical.

CL reports personal fees from Chiesi, Gilead, Janssen, Lucane, Novartis, Oxoid, Berlin Chemie and Thermofisher outside the submitted work.

CH reports personal fees from Janssen outside the submitted work.

CJ, KIG and LK have no conflicting interests to declare.

This does not alter our adherence to PLOS ONE policies on sharing data and materials.

2) All details regarding the approval of the ethics committee were moved to the last paragraph of the Material and Methods section of the manuscript.

---

## [Decision Letter · Decision Letter 1]

21 Aug 2020

Non-invasive ventilation with pursed lips breathing mode for patients with COPD and hypercapnic respiratory failure: a retrospective analysis

PONE-D-20-16968R1

Dear Dr. Herzmann,

We’re pleased to inform you that your manuscript has been judged scientifically suitable for publication and will be formally accepted for publication once it meets all outstanding technical requirements.

Kind regards,

Davor Plavec, MD, MSc, PhD, Prof.

Academic Editor

PLOS ONE

Additional Editor Comments (optional):

Reviewers' comments:

Reviewer's Responses to Questions

**Comments to the Author**

1. If the authors have adequately addressed your comments raised in a previous round of review and you feel that this manuscript is now acceptable for publication, you may indicate that here to bypass the “Comments to the Author” section, enter your conflict of interest statement in the “Confidential to Editor” section, and submit your "Accept" recommendation.

Reviewer #1: All comments have been addressed

Reviewer #2: All comments have been addressed

2. Is the manuscript technically sound, and do the data support the conclusions?

Reviewer #1: Yes

Reviewer #2: Yes

3. Has the statistical analysis been performed appropriately and rigorously? 

Reviewer #1: I Don't Know

Reviewer #2: I Don't Know

4. Have the authors made all data underlying the findings in their manuscript fully available?

Reviewer #1: Yes

Reviewer #2: Yes

5. Is the manuscript presented in an intelligible fashion and written in standard English?

Reviewer #1: Yes

Reviewer #2: Yes

6. Review Comments to the Author

Reviewer #1: Explanations about some unexpected results previously mentioned in my review are now very good addressed and have logic pathophysiology rational.

Reviewer #2: (No Response)

7. PLOS authors have the option to publish the peer review history of their article (what does this mean?). If published, this will include your full peer review and any attached files.

Reviewer #1: No

Reviewer #2: No

---

## [Editor Report · Acceptance letter]

11 Sep 2020

PONE-D-20-16968R1

Non-invasive ventilation with pursed lips breathing mode for patients with COPD and hypercapnic respiratory failure: a retrospective analysis

Dear Dr. Herzmann:

I'm pleased to inform you that your manuscript has been deemed suitable for publication in PLOS ONE. Congratulations! Your manuscript is now with our production department.

Kind regards,

on behalf of

Dr. Davor Plavec 

Academic Editor

PLOS ONE